# Soluble Plasma Proteins of Tumor Necrosis Factor and Immunoglobulin Superfamilies Reveal New Insights into Immune Regulation in People with HIV and Opioid Use Disorder

**DOI:** 10.3390/vaccines12050520

**Published:** 2024-05-09

**Authors:** Priya P. Ghanta, Christine M. Dang, C. Mindy Nelson, Daniel J. Feaster, David W. Forrest, Hansel Tookes, Rajendra N. Pahwa, Suresh Pallikkuth, Savita G. Pahwa

**Affiliations:** 1Department of Medicine, Division of Infectious Diseases, University of Miami Miller School of Medicine, Miami, FL 33136, USA; priyaghanta305@gmail.com (P.P.G.); dforrest@med.miami.edu (D.W.F.); hetookes@med.miami.edu (H.T.); 2Department of Microbiology and Immunology, University of Miami Miller School of Medicine, Miami, FL 33136, USA; cxd624@med.miami.edu (C.M.D.); rxp240@med.miami.edu (R.N.P.); spallikkuth@med.miami.edu (S.P.); 3Department of Public Health Sciences, University of Miami Miller School of Medicine, Miami, FL 33136, USA; m.nelson4@med.miami.edu (C.M.N.); dfeaster@med.miami.edu (D.J.F.)

**Keywords:** opioid use disorder, HIV and opioids, TNF superfamily and immunity, immunoglobulin superfamily, opioids and flu vaccine

## Abstract

People with HIV (PWH) frequently suffer from Opioid (OP) Use Disorder (OUD). In an investigation of the impact of OUD on underlying immune dysfunction in PWH, we previously reported that OP use exacerbates inflammation in virally controlled PWH followed in the Infectious Diseases Elimination Act (IDEA) Syringe Services Program (SSP). Unexpectedly, Flu vaccination-induced antibody responses in groups with OUD were superior to PWH without OUD. Here, we investigated the profile of 48 plasma biomarkers comprised of TNF and Ig superfamily (SF) molecules known to impact interactions between T and B cells in 209 participants divided into four groups: (1) HIV+OP+, (2) HIV−OP+, (3) HIV+OP−, and (4) HIV−OP−. The differential expression of the top eight molecules ranked by median values in individual Groups 1–3 in comparison to Group 4 was highly significant. Both OP+ groups 1 and 2 had higher co-stimulatory TNF SF molecules, including 4-1BB, OX-40, CD40, CD30, and 4-1BBL, which were found to positively correlate with Flu Ab titers. In contrast, HIV+OP− exhibited a profile dominant in Ig SF molecules, including PDL-2, CTLA-4, and Perforin, with PDL-2 showing a negative correlation with Flu vaccine titers. These findings are relevant to vaccine development in the fields of HIV and OUD.

## 1. Introduction

Opioids are a drug class of narcotic painkillers of varying intensity. Medically, opioids are used for the relief of severe pain; their analgesic effect is mediated through opioid receptors in the brain. Opioid Use Disorder (OUD) is characterized by dependence on the use of opioids despite experiencing adverse consequences [1] and includes persons who inject drugs (PWID). About 11 million people globally are PWID. Injection drug use is a risk factor for infections such as the Human Immune deficiency Virus (HIV), hepatitis B, hepatitis C, and other bacterial pathogens that result in infective endocarditis [2]. Among various opioids, the drug fentanyl is the most common in the street supply and is 50 times more potent than heroin [1]. The most dangerous complication of OUD is opioid overdose, which can cause death from respiratory depression. According to the Centers for Disease Control (CDC), 106,699 drug overdose deaths were reported in the United States in 2021 [3].

The prevalence of substance use disorders, including OUD, is high in people with HIV (PWH), and about one in eight inject drugs (UNODC World Drug Report, 2020) [4]. The Infectious Diseases Elimination Act (IDEA) Syringe Services Program (SSP) at the University of Miami (UM) that provides clinic and syringe exchange services to people with OUD, including those who have HIV, and embraces a harm reduction approach. Wraparound services include overdose education and naloxone distribution, medications for opioid use disorder as well as HIV/HCV prevention, screening and treatment... Even on ART, PWH have some level of immune dysfunction, consisting of inflammation and immune activation (IA) [5]. The mechanism underlying the inflammation and IA in PWH is not well understood, but it is speculated that the HIV reservoir may not be completely inert. Excessive inflammation and IA are major contributors to many comorbidities in PWH [6], examples of which include coronary artery disease, neurocognitive impairment, cancers, and diabetes. Inflammation is also considered to be a cause of inadequate immune responses to vaccines [5,7,8]. Inflammation is a multifactorial process and many different diseases are attributed to increased inflammation. Therefore, understanding and targeting inflammation and HIV reservoirs are active areas of research.

To understand the impact of Opioid use in PWH, we are conducting a study called OPIS (Opioid Immunity study). The overall goal of OPIS is to understand how OUD in PWH affects the body’s immune system. OPIS is enrolling participants from the SSP who have OUD with or without HIV, as well as PWH without OUD from the University of Miami Infectious Diseases clinics. We consider our “Control sample” to be those participants with neither HIV nor OUD. The rationale for OPIS is that opioids have a high likelihood of affecting the immune system because opioid receptors are also expressed on various immune cells, an understudied area, particularly in PWH. In animal models, chronic opioid use increases inflammation [9]. PWH also have inflammation and IA despite virus suppression with ART. In OPIS, we hypothesized that PWH who have OUD would have excessive inflammation and IA, more than that caused by HIV or opioids alone. Our recent study investigated 20 cytokines as well as cellular markers of activation and exhaustion [10]. In support of our hypothesis, we found that PWH who had OUD (HIV+OP+) exhibited the highest inflammatory cytokines and cellular IA, greater than that in PWH without OUD (HIV+OP−). Opioid use was the primary driver of inflammation, while living with HIV promoted IA [10].

Our second hypothesis in OPIS was that PWH and OUD would have deficiencies in immunity that would surpass those of PWH who did not have OUD. We evaluated immune competence by determining the magnitude of antibody response to the seasonal influenza (Flu) vaccine. In previous studies of Flu vaccines, investigations in our group noted that PWH have deficiencies in antibody responses to Flu vaccines despite virus suppression by ART [7,8,11,12] and that this impairment was linked to excessive inflammation and IA. Unexpectedly, instead of being deficient, overall antibody responses to the Flu vaccine in PWH plus OUD were far better than responses in PWH without OUD, who had the poorest responses, as expected. People with OUD without HIV also showed strong antibody responses (Manuscript in preparation). These findings suggested that despite excessive inflammation in people with OUD, opioids were augmenting immunity and overcoming deficits associated with HIV. The present study was conducted to test the hypothesis that chronic opioid intake leads to immunologic effects that can compensate for immune deficiencies associated with HIV as evaluated by antibody production in response to Flu vaccination. Here, we focused on the TNF and Ig superfamily molecules because several of their molecules are critical in regulating T helper cell function, which stimulates B cell maturation and antibody production [13]. These molecules are expressed in particular immune cells when they are activated and can be shed or secreted into the circulation [13,14], where they can be measured. In the cancer field, they are commonly used as “liquid biopsies” for diagnosis and monitoring of patients undergoing chemotherapy or radiotherapy. The results of the current study may point to a novel focus of research in the fields of opioid substance abuse disorder and vaccinology.

## 2. Materials and Methods

Study participants: The participants for the OPIS study were recruited as described [10] by the study recruiters (Appendix A). They were divided into four study groups based on their HIV status (HIV+/−) and OUD (OP+/−) status as HIV+OP+, HIV−OP+, HIV+OP−, and HIV−OP−. Both OP+ groups were recruited from the IDEA clinic; the HIV+OP− group was recruited from the Infectious Disease clinic, whereas the people without HIV or OUD were recruited from the community. All the people with HIV were required to have been on antiretroviral therapy (ART) for 6 months or longer with plasma HIV RNA levels less than 200 copies/mL to avoid confounding effects of the virus. People using opioids were required to have been using them for 90 days or more. Urine drug screening at study visits was performed at the SSP using point of care competitive binding immunoassays. Other participants for OP− groups were also tested for drugs and to confirm negative opioid status. Clinical data and responses to questionnaires are stored in a database (REDCap). All study participants were given a single intramuscular (i.m.) vaccine dose of quadrivalent influenza vaccination (QIV) containing 15 μg hemagglutinin of each vaccine antigen. This study was approved by the University of Miami Institutional Review Board (IRB#20200178), and participants were enrolled after obtaining signed informed consent. All participants had blood drawn at study entry at pre-vaccination (T0) and at day 7 (T1), day 21–28 (T2), and month 5–7 (T3) post-vaccination. Blood samples were processed within 2 h for serum and plasma and were stored at −80 °C. Participant characteristics are summarized in Table 1. For this study, 209 participants ranging in age from 24 to 68 years were evaluated. The PWH were on ART and mostly had viral suppression (HIV RNA < 200 copies/mL) and CD4 counts > 500 cells/uL.

Antibody titers to Flu vaccines at pre- and post-vaccination were determined to identify responders and non-responders in a subset of participants based on the hemagglutination inhibition (HAI) assay [12]. For each participant, a Flu vaccine score was calculated based on the sum of the log2 transformed fold change (T2/T0) responses to each of the four Flu antigens that year = Σ (log__2_(T2/T0), H1N1 + log__2_(T2/T0), H3N2 + log__2_ (T2/T0), B1 + log__2_ (T2/T0), B2) [15,16]. Vaccine responders are defined by a vaccine score greater than 4, while people with a vaccine score of less than or equal to 4 were classified as vaccine non-responders as described previously [7,8,12].

Plasma biomarker analysis for TNF and Ig superfamily: Plasma was collected from blood by centrifugation at 1000× *g* for 15 min and stored frozen at −80° in multiple aliquots for different studies. For the proteomic studies, 48 superfamily molecules, shown in Appendix A, were analyzed using flow-based bead Luminex assays [5,17] with Millipore Milliplex Checkpoint Protein Panel 1 for 17 proteins and Panel 2 for 31 proteins according to manufacturer instructions. Briefly, the microparticles treated with plasma underwent wash steps and staining with biotinylated antibody against the proteins, captured by Streptavidin–phycoerythrin, resuspended in wash buffer, and acquired on a Luminex Flexmap3D instrument for determining mean Fluorescence intensity. Protein concentrations were determined based on standard curves and expressed in pg/mL.

Statistical Analysis: Plasma protein expression (normalized individual protein marker expression and superfamily scores) was compared between groups. For each marker, we calculated the median expression level in the control group (Group 4: HIV−OP−). Markers of all participants were normalized by log _2_ transformation of the ratio of Marker expression of each participant/Median Healthy Control Marker expression. A normalized TNF and Ig superfamily heatmap was made with subjects ordered by group and hierarchical clustering of the molecules per subject. TNF and Ig Superfamily scores were calculated by summing all of the normalized Superfamily values divided by their number. Thus, the Superfamily scores represent the mean log-normalized expression values with respect to the median TNF Superfamily or Ig superfamily expression in the control group (group 4) and are depicted as violin plots for each group. Group differences in Superfamily scores were compared with a non-parametric Kruskal-Wallis Test followed by pairwise Dunn’s tests. In order to examine whether CD4 impacts the group differences in Superfamily score, a linear regression was conducted to look at both the association of CD4 and Group on TNF Superfamily score, with Group 4 as the reference level.

For group comparisons of individual markers, we performed non-parametric Kruskal–Wallis Tests followed by pairwise Dunn’s tests, correcting for multiple testing among groups using the Benjamini and Hochberg approach (v2022.12.0 RStudio). We further adjusted p values for the 48 separate analyses of the 48 markers and ranked by significant differences between groups. The top 8 molecules were then ranked by the transformed expressions of median values of individual molecules in Groups 1–3 in comparison to Group 4 and shown in Table 2. Spearman Rank Correlation analysis was used to examine the relationships between the normalized 48 TNF and Ig SF protein markers at baseline (time of vaccination) and Flu antibody titers at 28 days post-vaccination (T2) and Vaccine Score (v9.2.0 GraphPad Prism Inc., San Diego, CA, USA). Counts for Flu vaccine responders and non-responders in the 4 study groups were compared with Chi-squared tests followed by Fisher’s exact pairwise tests.

## 3. Results

### 3.1. TNF and Ig Superfamily Molecules

We examined 48 analytes of the TNF and Ig superfamilies. Figure 1 shows a composite heat map for visualization of the expression of the superfamily molecules analyzed in the four groups, showing heterogeneity in their distribution. The present study took a cue from the results of our previous study [10], in which we analyzed 20 soluble plasma cytokines that play a role in inflammation. In that study, TNFR1, TNFR2, and TNF-α were among the top five most elevated molecules identified in the HIV+ and OP+ groups (Appendix A, adapted from ref [10]. We analyzed the scores of all the superfamily molecules for each participant group (Figure 2). Collectively, when all 48 molecules were considered, the superfamily molecule score was greater in Groups 1–3 in comparison to Group 4. Scores ranged from −0.94 to 1.89, with a median of 0.57, 0.49, 0.27, and −0.03 for Groups 1–4, respectively, with Group 1 (HIV+OP+) having the highest value. Even while controlling for CD4, Group 1 (SE = 0.13, *p* < 0.00004), Group 2 (SE = 0.12, *p* < 0.002), and Group 3 (SE = 0.11, *p* < 0.003) were significantly different from Group 4 in TNF Superfamily score (TNF Superfamily Score = −0.13 + 0.00008 × CD4 + 0.55 × Group1 + 0.37 × Group 2 + 0.33 × Group3; Multiple R^2^ = 0.13, Adjusted R^2^ = 0.11, F_4,134_ = 5.08, *p* < 0.0008). CD4 was not significant (SE = 0.00009, *p* < 0.41). The analysis met distributional assumptions for a linear model.

To determine which molecules were different between groups, we examined the expression of each of the 48 molecules in the four groups. Figure 3 shows the molecules with the greatest inter-group differences. Plasma molecules 4-1BB, 4-1BBL, OX-40, CD40, CD40L, CD30, Nectin-2, and TIM 3 were higher in Groups 1 or 2 or both in comparison with Group 3 and Group 4. PDL-2, CTLA-4, BAFF, Granulysin, Perforin, GITR, Nectin-2, TIM 3, HVEM, Galectin, and CD27 were significantly higher in Group 3 than in Group 4. The other 24 molecules were comparable across all the four groups.

To arrive at a definitive distribution of these soluble molecules, we determined the top eight molecules with the highest transformed median values of the TNF and Ig superfamily in each of the study Groups 1–3 relative to Group 4 (Table 2). These molecules in Groups 1–3 were also highly significant in their difference from Group 4. In Opioid Groups 1 (HIV+OP+) and 2 (HIV−OP+), the majority (5/8) consisted of TNF SF molecules, while in Group 3, the Ig SF molecules were dominant (4/8). The most highly ranked TNFSF molecules shared only in Groups 1 and 2 were OX40, CD40, and 4-1BB, while CD30 and 4-1 BBL were distinctive to Group 2. In Group 3 (HIV+OP−), the distinctive highly ranked Ig SF molecules were CTLA-4, Perforin, and PDL-2. TIM-3 was common in all three groups. The HIV+ groups 1 and 3 shared the TNF SF molecules CD27 and HVEM among the top eight molecules of each group.

### 3.2. Correlation of TNF and Ig Superfamily Molecules with Flu Ab Titers

Group differences in vaccine responses are depicted in Appendix A. Group 3 had significantly lower vaccine scores than the other three groups (Appendix A), and the ratio of vaccine responders to non-responders was also significantly lower in Group 3 than in the other groups (Appendix A). A correlation matrix of plasma levels of different molecules at baseline (just prior to Flu vaccination) with the antibody response at 28 days post-vaccination is depicted in Figure 4. We picked the molecules with the most significance (*p* < 0.01) and plotted them in Figure 4B, which shows that a positive correlation with CD40, CD40L, 4-1BB, 4-1BBL, OX-40, and CD30 from TNFSF and with Tim3, Nectin 2, and Siglec-7 from Ig SF. PDL-2, also of Ig SF, was negatively correlated. A negative correlation of Perforin (not shown in Figure 4B) was found only for H1N1 response, indicating variability of molecules that correlate with responses to different antigens in the Flu vaccine. IDO1, not a member of the SF, correlated positively with vaccine score for the whole vaccine, H3N2, B1, and B2 antigens.

## 4. Discussion

In this study, we focused our attention on the co-stimulatory molecules of the TNF superfamily members, which are known to be important for Ab responses [11,12,15,16]. We also investigated proteins of the Ig superfamily to obtain a broad view of the immunoregulatory landscape [13,18,19,20,21,22]. We report for the first time that chronic opioid use in people with or without HIV results in a predominance of TNF SF molecules in plasma that correlate positively with Flu vaccine responses. On the other hand, PWH who do not use opioids show a predominance of inhibitory molecules of the Ig SF like PD-L2 and Perforin that correlated negatively with Flu vaccine responses. Although the underlying mechanism is unclear, upregulation of a distinct profile of the plasma molecules of the TNF SF by chronic opioid use in PWH appears to overcome the immunologic effects of HIV that lead to impaired antibody response to Flu vaccination. This is the first report on the induction of TNF SF molecules in people with OUD that warrants further investigation of the molecular pathways to identify biomarkers and therapeutic targets that could benefit immune-compromised states [12].

Our study consisted of circulating plasma molecules of receptors and ligands of the TNF and Ig SF. Membrane-bound and intracellular molecules can be shed into the circulation by proteolytic cleavage or may be secreted by particular cells. For example, to reduce inflammation, the ADAM17 enzyme cleaves the receptors and ligands from membranes into the plasma [23]. Overactivation of T cell immunity may be controlled by soluble molecules in an autoregulatory feedback mechanism in certain situations. Interaction between receptors and ligands can occur via molecules on cell membranes or by interaction between soluble plasma molecules with cell-associated molecules [24,25,26,27,28,29]. Measuring specific molecules in the plasma can determine if cells in certain superfamily pathways are active. We analyzed 48 soluble molecules predominantly of the TNF and Ig SF consisting of stimulatory and inhibitory immune receptors and their natural ligands, also known as immune checkpoints. Stimulatory immune checkpoint molecules boost the magnitude of cell activation initiated by the engagement of T cell receptors (TCR)/B cell receptors (BCR), whereas inhibitory immune checkpoints negatively regulate immune cell activation. Both pathways play important roles in mediating effective immune responses that control immune activation, and their dysregulation can lead to infection, cancer, autoimmunity, diabetes, and heart disease [24,26,27,28] The interaction of different cells with different molecules leads to the activation of different pathways. Some key membrane-bound TNF molecules are known to regulate human immune responses [13]. Examples include the interaction of TNFRs and TNF for T cells, monocytes, and DC, the interaction of 4-1BB and 4-1BBL for T cells and DC, the interaction of OX40 and OX40L for T cells, DC, and B cells, and the interaction of CD30 with CD30L, and CD40 with CD40L for T cells and DC; others include GITR and GITRL for T cells and stromal cells; HVEM and CD160, BTLA, LIGHT, and LTa3 for T cells, DC, Granulocytes, monocytes, and B cells. The molecules CD40, 4-1BB, OX40, and CD30 are members of the TNF SF and play important co-stimulatory roles in T cell activation [13,19,21]. The four pathways involving these molecules are especially relevant in the context of vaccine responses [20,28,29,30].

Collectively, when all 48 molecules were considered, the superfamily molecule score was greater in Groups 1–3 in comparison to Group 4. Individual molecule comparisons revealed differences between the groups for 24 of 48 molecules, and the individual value and significance of these molecules were ranked in comparison to Group 4. The 24 molecules were also analyzed for correlation to vaccine responses. When compared to control Group 4, the most highly ranked eight molecules in each group revealed that plasma molecules in HIV+OP+ Group 1 and HIV−OP+ Group 2 were more similar to each other and different from the HIV+OP− Group 3. Importantly, several of these molecules were positively correlated with Flu vaccine-induced antibody responses. Among the two OUD groups, the eight most significant proteins were CD40, OX-40, 4-1BB, CD 30, and 4-1BBL of the TNF SF, indicating a strong interaction between DC, T cells, and B cells. In contrast, in the HIV+ OP− Group 3, the unique Ig SF molecules compared to Group 4 were CTLA-4, Perforin, and PDL-2, of which PDL-2 and Perforin were negatively correlated with Ab responses, while CTLA-4 showed a trend. The TNF SF molecules that were common to the two HIV groups were CD137, CD40, and OX 40, and correlated positively with Flu Ab responses. TIM 3 of the Ig SF was common to all three groups and also correlated positively with antibody responses. Overall, these findings imply that the Ig SF molecules trending with poorer response to the Flu vaccine were prominent in the HIV+ OP− Group, while specific TNF SF molecules upregulated by Opioids in Groups 1 and 2 had a positive effect on antibody responses. These molecules need to be further investigated as they could reflect an outcome of DC, NK cell, and T cell interaction [31,32].

Among TNF SF molecules, CD40 in Group 1 was higher than Groups 3 and 4, and in Group 2, it was higher than Group 4 in quantitative analysis, and medians ranked in the top eight molecules in both groups. In quantitative analysis, CD40L was higher in Group 1 in comparison with Groups 3 and 4 and was equivalent to Group 2. CD40 and CD40L both correlated positively with Flu Ab responses. CD40 is a co-stimulatory cell surface receptor present on antigen-presenting cells (APC) such as DC, monocytes and macrophages, other myeloid cells, and B cells. It is essential for mediating a wide variety of immune and inflammatory responses, including T cell dependent signaling to B cells in immunoglobulin (Ig) isotype switching, somatic hypermutation of the immunoglobulins to enhance affinity for antigen and formation of long-lived B cells and plasma cells and promoting GC formation [33,34]. CD40L expression on CD4 and CD8 lymphocytes occurs shortly after T cell activation [35]. CD40L-mediated activation of monocytes and macrophages and DC that express CD40 leads to the production of IFNg, IL-12, IL-1b, IL-6, IL-8, and TNF. In the opioid groups 1 and 2, activation of the CD40 pathway in Flu vaccine responders indicates increased T cell priming and functional activity of T-dependent immunoglobulin class switching, memory B cell development, and long-lived plasma cell formation.

Like CD40, 4-1BB, and 4-1BBL of the TNF SF were elevated in Groups 1 and 2, ranked in the top eight molecules in one or the other group, and correlated positively with Flu Ab responses. The co-stimulatory molecule 4-1BB has roles in expansion, acquisition of effector function, survival, and development of T cell memory [36,37,38,39]. The 4-1BB ligand (4-1BBL) is expressed on dendritic cells (DCs), while its receptor is found on T cells. Additionally, 4-1BB is expressed on activated T cells, endothelial, and epithelial cells [40]. The 4-1BBL is also expressed on antigen-presenting cells (APCs). When 4-1BBL binds with 4-1BB, it induces signaling through the TRAF1 and TRAF2 pathways, activating the NF kB, AKT, p38 MAPK, and ERK pathways. Together, these pathways encode survivin, Bcl-2, Bcl-XL, and Bfl-1 [40,41]. These pathways help the cells survive and proliferate, and they also decrease the expression of the pro-apoptotic gene, Bim [41]. Our findings indicate that the important 4-1BB pathway was stimulated with opioid use both in people with and without HIV.

OX-40 of the TNF SF was highly expressed in OP+ Groups 1 and 2 in comparison to Groups 3 and 4, was among the top-ranked molecules in these groups, and correlated positively with Ab responses. OX-40 provides a co-stimulatory signal to activated T cells during its transient expression through interaction with OX-40L following TCR ligation [40,41,42,43], which is the first signal. OX-40 mediates its effects through TRAF-2 and TRAF-5 to activate canonical and non-canonical NF-kB, PI3K, and AKT [44,45,46,47]. OX-40 ligation on activated T cells leads to enhanced proliferation and development of effector functions, survival of memory T and B cells, and skewing of the CD4 population toward TH-2 cell phenotype, leading to secretion of IL-4, a cytokine that helps in B cell differentiation and antibody formation [40,41,47,48].

CD30 of the TNF SF was higher in Group 2 in comparison to HIV Group 3 and control Group 4 in quantitative analysis. It showed a trend to be higher in Group 1, and its levels correlated positively with Flu vaccine response. CD30 is expressed on activated T and B cells as well as on subsets of memory and Treg cells [47,49]. CD30 ligand on professional APCs induces either cell death or cell-cycle arrest or survival and proliferation of CD30+ cells. CD30 utilizes TRAF 2 and TRAF 3 to activate a variety of pathways, including NFkB, p38 MAPK, ERK, AKT, and AP-1 [44,50,51,52,53,54,55]. In models of bacterial infection, genetic deletion of CD30 or CD 30L leads to a decrease in the long-term ability to control the pathogen that is associated with the loss of long-lived CD4+ T cells secreting IFNg and defective CD8+ T cell memory cells.

It should be noted that TNFR1, TNFR2, and TNF-α were also among the most significant cytokines in our initial analysis for inflammatory cytokines [10] and were highest in Group 1, followed by Group 2, being significantly higher in Groups 1 and 2 than Group 3 and control Group 4. TNF-α is cleaved by ADAM17, which also cleaves TNFR2 during activation of effector T cells. TNFR2 is expressed on B and T cells, Treg cells, DCs, macrophages, neural cells, and endothelial cells. The binding of TNF to TNFR2 on DC activates them to undergo maturation. TNFR2 in activated T cells provides a co-stimulatory signal required for T cell proliferation and effector differentiation. The activation of TNFR2 is important for the proliferation, survival, and lineage stability of Treg cells and the development of thymic Treg cells, leading to an increase in sCD25, as was noted in Groups 1, 2, and 3. The activation of TNFR2 triggers the pro-survival NF-κB pathway via E3 ligases TRAF2 and TRAF3 [56]. CD25/IL2Ra was significantly greater in Groups 1–3 compared to Group 4, indicating that Tregs were activated in all three groups.

A question that arises is how opioids were causing activation of the SF molecules that were being shed off or secreted and what is the underlying mechanism of opioid-mediated boosting of the immune response. We can speculate on at least two mechanisms. One is the interaction of opioids with opioid receptor-expressing immune cells. There are three main types of opioid receptors on immune cells: μ (MOR), δ (DOR), κ (KOR), for which opioids have agonistic activity. All opioid receptors are expressed on immune cells [57]. OP/OPR signaling effects on immune cells are controversial but may induce the secretion of specific molecules, including TNF SF and inflammatory cytokines. The observed opioid effects could be direct or indirect through the induction of inflammatory cytokines, as already demonstrated [10]. Another potential mechanism for innate cell activation could be pathogens from the skin entering the bloodstream through the process of injection drug use.

In vivo innate cell activation may play an important role in the generation of soluble plasma protein profiles seen in people with OUD and their downstream effects. Under steady-state conditions, DC remains in an immature state, unable to initiate effector T cell responses, and can induce T cell tolerance. In response to danger signals (e.g., pathogens from injecting drugs, virus antigens), DC expresses enhanced levels of MHC II peptide complex and co-stimulatory molecules and secrete cytokines that drive distinct T cell responses, e.g., into TH1, TH2, TH17, or T reg, that depend on the nature of signals [58,59]. It is likely that in people with OUD (Groups 1 and 2), DC and T cells are already activated at a steady state and primed prior to exposure to Flu vaccine antigens, resulting in a boosted Flu-specific immune response. Interaction of activated DC with T cells leads to increased activation of Flu vaccine antigen-specific T cells through co-stimulatory pathways, resulting in stimulation of B cells, leading to Ig class switching, antibody production, generation of memory B cells, and long-lived Flu-specific plasma cells [33,34]. Increases in plasma levels of soluble proteins such as 4-1BB, 4-1 BBL, CD40, OX40, CD30, TNF, TNFR 1, and TNFR2 are indicative of innate cell activation. Dendritic cell maturation and survival are regulated by TNF, TNFR1, TNFR2, CD40, and 4-1BB. It is well known that TNF and its receptors TNFR1 and TNFR2 promote both DC maturation and survival [19,60]. The survival of DC is regulated by several TNF superfamily members that can either work independently, cross-react and/or cooperate with TNFR signaling [61,62]; 4-1BB and CD40 help in DC survival and maturation [63]; 4-1BB and TNFRs are particularly potent in promoting survival through upregulation of BCL-2 and/or BCL-XL for survival and CD40 signaling results in full DC maturation [29,60].

Soluble forms of receptors and ligands of TNF and Ig superfamily molecules, namely CD30, OX40, OX40L, CD40, CD30, CD95, GITR, GITRL, CTLA4, Tim3, Nectin-2, Siglec-7, PDL-1, and PDL-2 [24,25,26,27,28,29,64] have been reported in plasma in autoimmunity and a variety of cancers. An increase in plasma CTLA-4, a potent inhibitory checkpoint molecule, is a general phenomenon in cancer [27,28,29,64], and its transcripts have been detected in lymph nodes and in the spleen but not in non-lymphoid tissue. Soluble molecules, particularly CD40/CCD40L, CTLA-4/CTLA-4R, OX-40, 4-1BB/4-1BBL, and PD-1, are being investigated as diagnostics as well as for their therapeutic potential acting either as agnostic or antagonistic molecules [30].

## 5. Conclusions

We show for the first time that chronic opioid intake upregulates a distinct profile of the plasma molecules of the TNF SF that are linked to immune cell interactions between DC, T cells, and B cells, which are key determinants of antibody responses to Flu vaccines. Main molecules and facilitatory pathways in TNF SF are comprised of CD40, CD40L, 4-1BB 4-1BBL, OX40, and CD30, along with TNF and its receptors TNFR1 and TNFR2 for DC and T cell interaction, contributing to the formation of long-lived plasma cells and memory B cells. We need to understand if the SF proteins in plasma result from opioid/opioid receptor interaction in immune cells or some other pathway of cell activation. Whether opioids can act directly on the T cells and B cells to stimulate antibody production and the full action of different soluble molecules of TNF and Ig SF in plasma on immune function needs investigation. Different plasma SF molecules observed in people with OUD may hold clues for augmenting immunity, reversing exhaustion in T cells, or suppressing immunity when the immune system is overactive. Manipulation of the TNF SF molecules and ligands may present novel therapeutic approaches or strategies for vaccine adjuvants and in the treatment of OUD.

## Figures and Tables

**Figure 1 vaccines-12-00520-f001:**
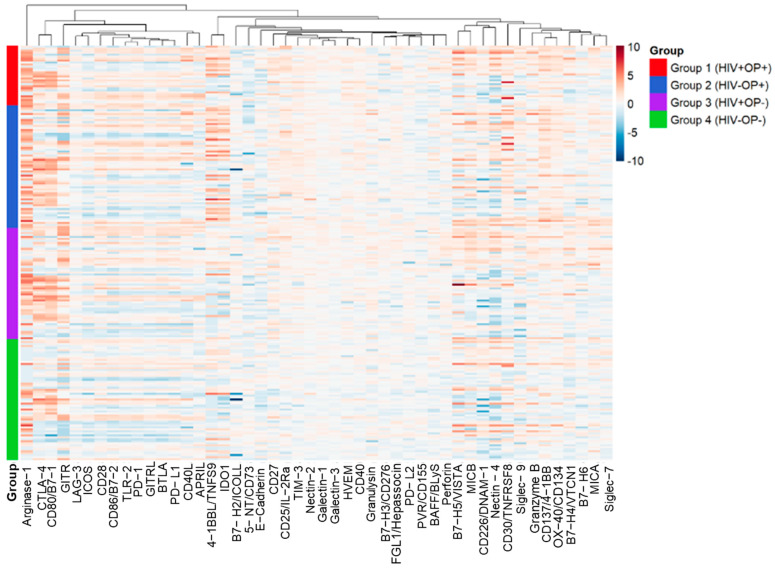
Heatmap of TNF and Ig Superfamily molecules in the 4 study groups. Normalized TNF and Ig superfamily heatmap with subject ordered by group and hierarchical clustering of the molecules. Each molecule was compared against the median expression level in the healthy control group (Group 4: HIV−OP−) and calculated as the log 2 transformed ratio.

**Figure 2 vaccines-12-00520-f002:**
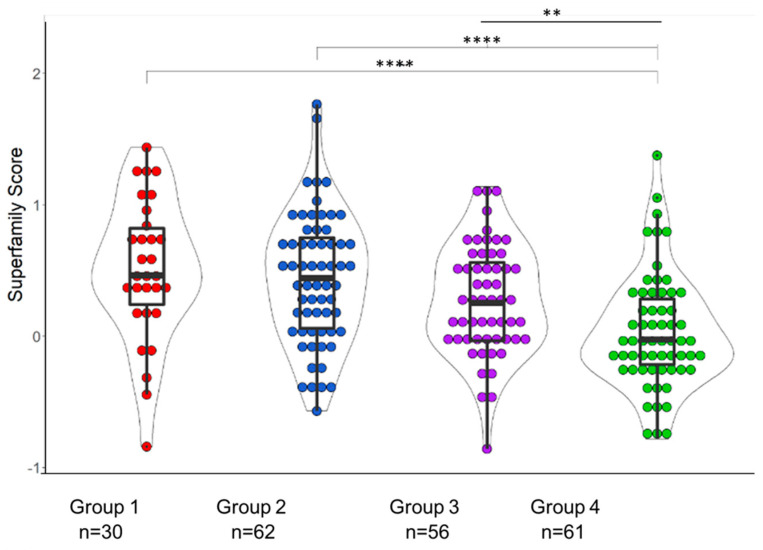
Composite Superfamily Score of 48 molecules. Using subject-level Superfamily scores, the 4 Groups were compared for their distributions using violin plots. The non-parametric Kruskal–Wallis test was corrected for multiple comparisons by controlling the FDR (original FDR method of Benjamini and Hochberg). Red represents HIV+OP+ (*n* = 30), Blue represents HIV−OP+ (*n* = 61), Purple represents HIV+OP− (n = 56), and Green represents HIV−OP− (n = 59). * indicates the statistical significance with adjusted *p*-values with number of * indicates the level of significance: **** *p* < 0.0001, ** *p* < 0.01.

**Figure 3 vaccines-12-00520-f003:**
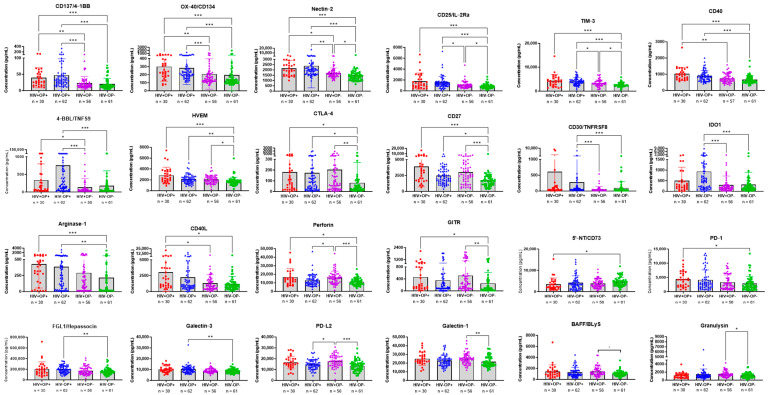
TNF and Ig Superfamily molecules that were significantly different among the 4 study groups. Levels of superfamily molecules with significant differences between groups: Red represents HIV+OP+ (*n* = 30), Blue represents HIV−OP+ (*n* = 61), Purple represents HIV+OP− (*n* = 56), and Green represents HIV−OP− (*n* = 59). The non-parametric Kruskal–Wallis test was corrected for multiple comparisons by controlling the FDR (original FDR method of Benjamini and Hochberg). * indicates the significant results and numbers of * indicate the level of significance with adjusted *p*-values: *** *p* < 0.001, ** *p* < 0.01, * *p* < 0.05.

**Figure 4 vaccines-12-00520-f004:**
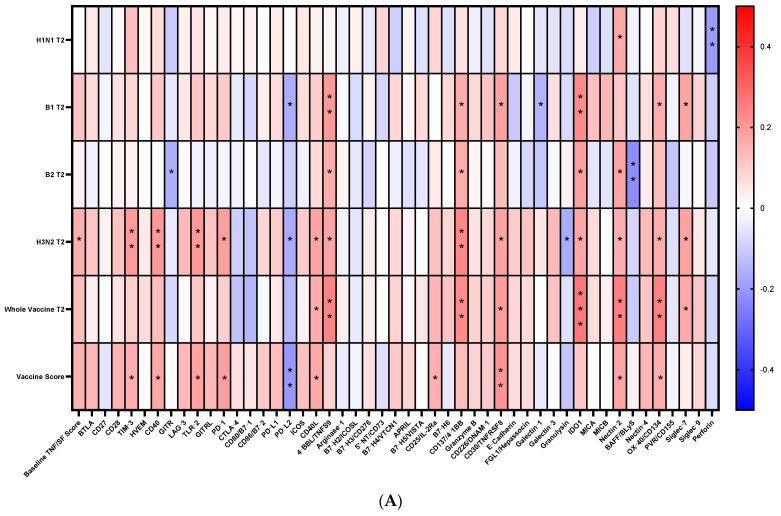
Correlation analyses (Spearman r) of plasma levels of all SF molecules at baseline (equivalent to time of Flu vaccination) with the antibody response at 28 days post-vaccination. (**A**) Heatmap showing correlations between baseline scores of superfamily molecules and individual superfamily molecules with the antibody titers against different Flu vaccine antigens; (**B**) Heatmap showing correlations of 11 selected SF molecules at T0 with Flu Ab titers at T2. *** *p* < 0.001, ** *p* < 0.01, * *p* < 0.05.

**Table 1 vaccines-12-00520-t001:** Participant demographics by Groups.

Population	HIV+OP+	HIV−OP+	HIV+OP−	HIV−OP−
Number	30	61	56	59
Median Age in Years (Range)	46 (28–63)	40 (24–63)	52 (32–68)	47 (24–62)
Gender: M/F/other (%)	33/67/0	25/75/0	46/48/5	49/51/0
Race: White/Black/Other (%)	73/17/10	77/20/3	36/63/1	31/63/6
Ethnicity: Hispanic/Non-Hispanic/Unknown (%)	43/53/4	28/70/2	30/70/0	36/64/0
Clinical information, median (IQR)				
CD4 T cell count (cells/uL)	685 (473–887)	872 (680–1040)	563 (367–937)	1106 (726–1344)
CD4/CD8 Ratio	0.70 (0.55–1.02)	1.67 (1.33–2.55)	0.75 (0.44–1.09)	2.20 (1.56–3.43)
HIV Viral Loads, RNA copies/mL *	24 (20–287)	NA	20 (20–100)	NA
Duration of ART, months	36 (24–48)	NA	192 (126–264)	NA

Total participants in the study were 209. Table shows 206 participants as demographics were not available for 3 participants, one in Group 2 and 2 in Group 3. CD4 T cell counts and CD4/CD8 Ratio were available for 66% of the participants (*n* = 23, 30, 53, 33 in Groups 1–4 respectively). * 20 is limit of detection; NA, Not Applicable.

**Table 2 vaccines-12-00520-t002:** Top 8 molecules ranked by median values for Groups 1–3.

	Comparison Groups		Numbers				
Molecules	Group 1 (HIV+OP+) Median (Q1, Q3)	Group 4 (HIV−OP−) Median (Q1, Q3)	N_1_	N_4_	Statistic	*p*<	*p* adj<
Arginase 1	3.714 (2.384, 4.357)	−0.101 (−0.898, 2.901)	30	60	−4.03	0.00006	0.002
CD137/41BB#	1.062 (0.388, 1.919)	−0.007 (−0.310, 0.629)	30	60	−4.37	0.00002	0.0002
CD27#	0.989 (0.108, 1.624)	0.002 (−0.621, 0.502)	30	60	−4.47	0.000008	0.0004
OX40/CD134#	0.901 (0.506, 1.618)	−0.035 (−0.355, 0.419)	30	60	−5.06	0.0000005	0.00003
CD25/IL2Ra	0.837 (0.042, 1.704)	−0.003 (−0.386, 0.385)	30	60	−4.15	0.00004	0.0007
TIM3 *	0.686 (0.122, 1.031)	−0.030 (−0.261, 0.270)	30	60	−4.55	0.000006	0.0002
CD40#	0.636 (0.278, 1.001)	−0.004 (−0.266, 0.346)	30	60	−5.13	0.0000003	0.00003
HVEM#	0.549 (0.202, 1.134)	−0.013 (−0.239, 0.229)	30	60	−4.69	0.000003	0.0002
Molecules	Group 2 (HIV−OP+) Median (Q1, Q3)	Group 4 (HIV−OP−) Median (Q1, Q3)	N_1_	N_4_	Statistic	*p*<	*p* adj<
CD137/41BB#	1.267 (0.657, 1.866)	−0.007 (−0.310, 0.629)	63	60	−6.12	0.000000001	0.00000007
4BBL/TNFSF#	1.241 (0.223, 3.201)	−0.119 (−1.191, 1.284)	63	60	−4.08	0.00005	0.0009
CD30/TNFRSF#	1.105 (0.354, 2.243)	0.058 (−0.725, 0.853)	63	60	−4.22	0.00003	0.0005
CD25/IL2Ra	0.860 (0.348, 1.154)	−0.003 (−0.386, 0.385)	63	60	−5.91	0.000000004	0.0000002
OX40/CD134#	0.755 (0.512, 1.217)	−0.035 (−0.355, 0.419)	63	60	−6.2	0.0000000006	0.00000007
TIM3 *	0.651 (0.336, 0.922)	−0.030 (−0.261, 0.270)	63	60	−6.16	0.0000000008	0.00000007
Nectin 2 *	0.524 (0.244, 0.737)	−0.012 (−0.311, 0.211)	63	60	−6.98	0.000000000004	0.0000000009
CD40#	0.483 (0.094, 0.805)	−0.004 (−0.266, 0.346)	63	60	−4.58	0.000005	0.0002
Molecules	Group 3 (HIV+OP−) Median (Q1, Q3)	Group 4 (HIV−OP−) Median (Q1, Q3)	N_1_	N_4_	Statistic	*p*<	*p* adj<
GITR#	1.449 (0.000, 2.519)	0.000 (−2.253,1.771)	56	60	−3.58	0.0004	0.03
CTLA 4 *	1.281 (−0.111, 2.832)	−0.042 (−0.815, 0.981)	56	60	−3.41	0.0007	0.03
CD27#	0.687 (−0.027, 1.389)	0.002(−0.621, 0.502)	56	60	−4.16	0.00004	0.003
Perforin *	0.497 (−0.029, 0.813)	0.005 (−0.404, 0.367)	56	60	−3.77	0.0002	0.02
CD25/IL2Ra	0.395 (0.018, 0.693)	−0.003 (−0.386, 0.385)	56	60	−2.88	0.004	0.04
PD_L2 *	0.348 (0.035, 0.683)	−0.045 (−0.446, 0.216)	56	60	−4.56	0.000006	0.002
TIM 3 *	0.298 (0.006, 0.689)	−0.030 (−0.261, 0.270)	56	60	−3.16	0.002	0.03
HVEM#	0.281 (−0.074, 0.646)	−0.013 (−0.239, 0.229)	56	60	−2.93	0.004	0.04

# indicates TNF-SF molecules and * indicates IG-SF molecules.

## Data Availability

The authors confirm that the data supporting the findings are available within the article and its Appendix A. Raw data that support the findings are available from the corresponding author upon reasonable request.

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
