# Peer review of "Soluble Plasma Proteins of Tumor Necrosis Factor and Immunoglobulin Superfamilies Reveal New Insights into Immune Regulation in People with HIV and Opioid Use Disorder"

_vaccines, 2024, doi:10.3390/vaccines12050520_

Round 1
Reviewer 1 Report
Comments and Suggestions for Authors
Soluble Plasma Proteins of Tumor Necrosis Factor and Immunoglobulin Superfamilies Reveal New Insights into Immune Regulation in People with HIV and Opioid Use Disorder.
In the present article, the authors have addressed the problem of Opioid (OP) Use Disorder (OUD) in People with HIV (PWH) who frequently suffer from this.
In this study, the authors have investigated the profile of 48 plasma biomarkers comprised of TNF and Ig superfamily (SF) molecules known to impact T-B interactions in 209 participants divided into four groups: (1) HIV+OP+, (2) HIV-OP+, (3) HIV+OP- and 21 (4) HIV-OP-.
The results of this study may inform research in the fields of opioid substance abuse disorder and vaccinology. The study is meticulously designed and well executed. The present manuscript is well written. The following are the specific comments to strengthen the current manuscript further,
1. In the abstract, the IDEA clinic appears suddenly without any detailed information; please consider providing full information with an abbreviation on its first appearance for ease of readers.
2. If this T-B interaction means interaction between T cells and B cells, it will be easy to follow if it can be mentioned as T and b cells.
3. In the introduction, line 84, “flu vaccine in people PWH plus OUD were far better than responses in PWH without OUD,” the underlined section, people may need to be removed from this sentence.
4. How can the vaccine be improved to fit OUD better?
Author Response
Reviewer 1.
We thank the reviewer for the favorable comments about the design and execution of the study and for the comments aimed to strengthen the manuscript further. Our responses to each comment are provided below:
- In the abstract, the IDEA clinic appears suddenly without any detailed information; please consider providing full information with an abbreviation on its first appearance for ease of readers.
We agree and have made the required change in the abstract. The IDEA clinic is further explained in lines 46 to 47 and again in lines 105-107 to explain participant recruitment..
- If this T-B interaction means interaction between T cells and B cells, it will be easy to follow if it can be mentioned as T and B cells.
We agree and have made the required change in the abstract in line 21.
- In the introduction, line 84, “flu vaccine in people PWH plus OUD were far better than responses in PWH without OUD,” the underlined section, people may need to be removed from this sentence.
We agree and have made the required change (line 85).
- How can the vaccine be improved to fit OUD better?
The people with OUD responded well to the vaccine and PWH with OUD (HIV+OP+) responded far better than PWH without OUD. This finding is relevant because PWH are poor vaccine responders in general, and chronic opioid use in PWH enhanced their Antibody responses to the influenza vaccine, overcoming negative factors contributing to deficient antibody responses in PWH. The findings in this manuscript are relevant for vaccinologists to gain insight into role of TNF SF molecules for enhancing vaccine immunogenicity in general. This issue is discussed in the Discussion and Conclusion section.
Reviewer 2 Report
Comments and Suggestions for Authors
In the present paper, the authors have investigated the profile of 48 plasma biomarkers of the TNF and Ig superfamily (SF) molecules in 209 participants divided into four groups: (1) HIV+OP+, (2) HIV-OP+, (3) HIV+OP- and (4) HIV-OP-. Both OP+ groups 1 and 2 had higher co-stimulatory TNF SF molecules, such as 4-1BB, OX-40, CD40, CD30 and 4-1BBL, which were found to positively correlate with Flu Ab titers. On the other hand, HIV+OP- showed a profile dominant in Ig SF molecules including PDL-2, CTLA-4 and Perforin, with PDL-2, with negative correlation with Flu vaccine titers.
The study is interesting and well performed. The text is clear with only minor editing required.
Suggestions for improvement:
-Figure 1 should not be put along M&M but combined along with Figure 2
-As shown in table 1, the CD4 count and the CD4/CD8 ratio is very different between HIV+ and HIV- patients. Since the study is focused on immune related molecules, the authors should perform the statistical analysis by correcting for the different levels of CD4 T cells, by using, for instance a GLM. If the results are not affected, they could just provide the results of this additional analysis as Suppl. Material.
Comments on the Quality of English LanguageMinor editing required
Author Response
We thank the reviewer for the positive comments about the study “The study is interesting and well performed. The text is clear with only minor editing required.”
Response to suggestions for improvement are included below;
- -Figure 1 should not be put along M&M but combined along with Figure 2.
We removed all reference to figures from M&M and have put them in Results, providing more clarity and ease of understanding. Positioning of figures is best done in the published draft, with the typesetting.
- -As shown in table 1, the CD4 count and the CD4/CD8 ratio is very different between HIV+ and HIV- patients. Since the study is focused on immune related molecules, the authors should perform the statistical analysis by correcting for the different levels of CD4 T cells, by using, for instance a GLM. If the results are not affected, they could just provide the results of this additional analysis as Suppl. Material.
The additional analyses performed are described in Methods (lines 182-184) and Results (lines 201-207).